# An Enhanced Deep Learning-Based DeepFake Video Detection and Classification System

Joseph Bamidele Awotunde [1], Rasheed Gbenga Jimoh [1], Agbotiname Lucky Imoize [2,3],
Akeem Tayo Abdulrazaq [1], Chun-Ta Li [4,5,*] and Cheng-Chi Lee [6,7]

1   Department of Computer Science, Faculty of Information and Communication Sciences, University of Ilorin, Ilorin 240003, Nigeria
2   Department of Electrical and Electronics Engineering, Faculty of Engineering, University of Lagos, Akoka, Lagos 100213, Nigeria
3   Department of Electrical Engineering and Information Technology, Institute of Digital Communication, Ruhr University, 44801 Bochum, Germany
4   Program of Artificial Intelligence and Information Security, Fu Jen Catholic University, New Taipei City 24206, Taiwan
5   Department of Information Management, Tainan University of Technology, Tainan City 71002, Taiwan
6   Research and Development Center for Physical Education, Health, and Information Technology, Department of Library and Information Science, Fu Jen Catholic Univesity, New Taipei City 24206, Taiwan
7   Department of Computer Science and Information Engineering, Asia University, Taichung City 41354, Taiwan
*   Correspondence: th0040@mail.tut.edu.tw

**Abstract:** The privacy of individuals and entire countries is currently threatened by the widespread use of face-swapping DeepFake models, which result in a sizable number of fake videos that seem extraordinarily genuine. Because DeepFake production tools have advanced so much and since so many researchers and businesses are interested in testing their limits, fake media is spreading like wildfire over the internet. Therefore, this study proposes five-layered convolutional neural networks (CNNs) for a DeepFake detection and classification model. The CNN enhanced with ReLU is used to extract features from these faces once the model has extracted the face region from video frames. To guarantee model accuracy while maintaining a suitable weight, a CNN enabled with ReLU model was used for the DeepFake-detection-influenced video. The performance evaluation of the proposed model was tested using Face2Face, and first-order motion DeepFake datasets. Experimental results revealed that the proposed model has an average prediction rate of 98% for DeepFake videos and 95% for Face2Face videos under actual network diffusion circumstances. When compared with systems such as Meso4, MesoInception4, Xception, EfficientNet-B0, and VGG16 which utilizes the convolutional neural network, the suggested model produced the best results with an accuracy rate of 86%.

**Keywords:** convolutional neural networks; DeepFake facial reconstruction; deep learning; DeepFake video detection; image alteration; generative adversarial networks; rectifying linear unit; recurrent neural networks

## 1. Introduction

Worry over news that is purposefully incorrect has increased, and artificial intelligence algorithms have recently made it simpler and more realistic to produce so-called "DeepFake" videos and images. These methods could be used to fabricate statements from well-known celebrities or videos of fabricated events, fooling large audiences in risky ways. The DeepFake network is a hotly debated topic in the field of security measures in various systems. Despite numerous recent developments in facial reconstruction, the hardest obstacle to solving this has been how to compute face similarity or matches in a timely and effective manner. Due to lossy compression and high data degradation, normal analysis techniques for detecting image forgery are often inappropriate for video forensics. Hence, due to restricted hardware and efficiency, real-time DeepFake facial reconstruction for

security purposes is challenging to complete. Therefore, this study proposes five-layered convolutional neural networks for DeepFake facial reconstruction and image segmentation. The performance evaluation of the proposed model was tested using various DeepFake datasets namely Face2Face, and first-order motion.

Artificially produced audio or visual renderings, most frequently videos, are known as DeepFakes. These videos, which are frequently made without the subject's consent, can be exploited to discredit important figures or sway public opinion. An audio or video recording might serve as uncontested evidence in a court of law. With generative adversarial networks (GANs), studied by the authors in [1,2], an attacker can create such accurate renderings by employing a standard desktop computer equipped with an aftermarket graphics processing unit. Both machines and people can be duped easily by them. In recent times, advanced DeepFake techniques used in face-based alteration have created an opportunity to substitute one person's face with another [3]. Thus, it appears incredible to make not only a copy–move modification but also to involve and implement supervised learning for automatically replacing one person's face with another. A clear set can now be animated and transformed into a sequence of video frames. Consequently, technology can now make even a statue come alive [4].

DeepFake replaces the facial features actively present, using a model known as the GAN, in actual footage with somebody else's face. GAN models have been developed while making use of several thousands of images, so this makes it attainable to create realistic faces which can be extracted and cropped into the original video in a manner that looks almost perfect. This resulting video can produce a higher authenticity via suitable post-processing or post-production processing [5]. The authors in [5] believed that before the advent of fake videos, videotapes were generally dependable and trustworthy, and this was in interactive media forensics, which is commonly used as hard evidence. The emergence of DeepFake videos, on the other hand, is eroding people's trust. There is growing concern that once this technology is used as proof in court, the media and publishing, diplomatic elections, and television and infotainment, it will be misused and have an impact on the lives of people which would be enormous. Some people even believe that this kind of technological advancement could mar the development of society. Therefore, identification and detection of such fake videos, either for official or non-official purposes, is cogently important.

As these manipulations become more persuading, public figures can be placed into an unreal scenario, consequently giving an impression that anybody could say anything you want them to say [6]. Even if the wider populace does not assume they are real, video evidence will become less reliable as a validation source, and this could also make the public lose their trust in whatever they see. This increases the urgency and strain on trusted hands in the mainstream media to help substantiate multimedia for general public consumption [6]. Several algorithms have been developed to detect DeepFakes, especially in videos. This study showed that while some of these methods have proven effective to some extent, most of these algorithms have also failed when evaluated with external data obtained from outside their study environments.

The privacy of people and nations is currently threatened by the widespread use of face-swapping DeepFake algorithms, which result in a sizable number of fake videos that are incredibly authentic. The ability to discern between DeepFake and actual films has become a crucial issue as a result of their harmful effects on society. The significant improvements in GANs and other methods of production have produced plausible false media that may have a very negative impact on society. On the other hand, the advancement of generation techniques is outpacing the effectiveness of the present DeepFake detection systems, thus resulting in a need for a better DeepFake detector that can be applied to media produced using any technique. The development of a DeepFake video/image detector can be generalized to various creative approaches that are presented within recent challenging datasets. Creating a system that will outperform the results produced by current state-of-

the-art methods using various performance measures, served as the underlying reason and motivation for this study.

In order to determine whether the target material has been edited or synthesized, DeepFake detection solutions typically employ multimodal detection techniques. Current detection methods frequently concentrate on creating AI-based algorithms for algorithmic detection techniques such as a Vision Transformer [7,8], MesoNet, which was suggested by the authors in [9], two-stream neural network [10], among others. Manual image processing, on the other hand, receives less consideration in favor of emphasizing the key areas of an image [11]. The model frequently becomes heavier as a result of processing all of the videos. In this study, we combined a human processing method with a DL-based model to enhance the DeepFake detection approach. Before being fed into a CNN-based model, the most crucial data, regions, and features are carefully selected and processed. Focusing on the most pertinent information helps these networks train more efficiently while also increasing the accuracy of the model as a whole.

To address these challenges, the proposed model has been trained against a large dataset of videos containing realistic manipulations and evaluated to ensure that the system works efficiently and effectively, and also, this is used to easily detect and classify a video as being a DeepFake or not. The main idea of the suggested model is to use a few of the most widely used classification models to recognize fraudulent videos and demonstrate how to reduce the complexity of the DeepFake detection challenge. Since the current classification models are built for high accuracy, judicious model selection will also improve the capacity to address the DeepFake detection issue. Therefore, this study aims to enhance the method of DeepFake detection using five-layered convolutional neural networks and image segmentation. The following are the main contributions of the proposed model.

i. Design a method of DeepFake video detection using CNN with a modified ReLU activation function to enhance the binary classification of DeepFakes on face-to-face and first-order motion datasets.
ii. To identify videos with high compression factors on the datasets, the performance of the optimized CNN algorithms and their modifications is experimentally demonstrated.
iii. Improving accuracy in low-resolution videos with less reliance on the number of frames needed for each video than current techniques, and evaluating the performance of the DeepFake detection system.

This paper is organized as follows. Section 2 gives a literature review of related works, Section 3 gives details about the methodology of the proposed system, Section 4 discusses the experiments, datasets used, classification and evaluation techniques used, and Section 5 presents the conclusion, recommendations for future improvement, and limitations of the proposed method.

## 2. Related Work

DeepFake codes are generated at the core using an autoencoder which is a deep neural network that studies and learns how to take an input, compress it down into a small representation or encoding, and then recreate the original input from this encoding [9]. Once the training of the dataset is complete, an image/video (source/original video) can be passed into the encoder. Then the place of trying to recreate the source from the encoding is passed to the decoder. Figure 1 displays a forged video creation process using the GAN model.

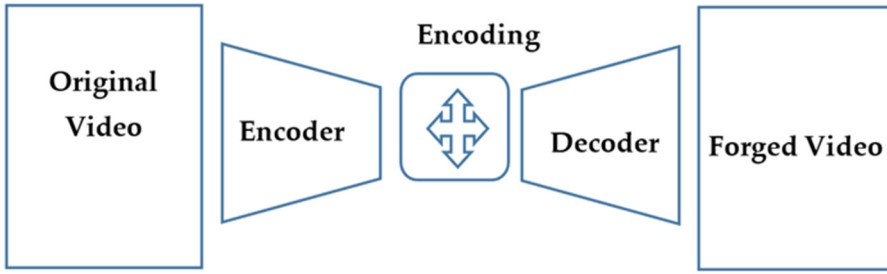

**Figure 1.** Forged video creation process using GAN architecture.

Various authors have been working on the identification of DeepFakes for the past few years. The early studies investigated visual irregularities and inconsistencies inside the frames. Some methods [12–14] focused on biological signals, whereas others [9,15] delegated feature extraction to convolutional neural networks (CNNs). The authors in [16] proposed to employ capsule networks with dynamic routing and obtain extremely good results. Several other methods were successful in localizing the altered regions (generally, the face) [17–19]. DeepFakes' capacity to disseminate false information that looks to emanate from international leaders is a serious worry, therefore, authors in [16] suggested tracking facial landmarks to learn behaviors typical of particular people and use them to distinguish between authentic and false video content.

Image alteration has been practiced for many years, but it is time- and money-consuming. Deep convolutional neural networks have recently advanced (CNNs), nonetheless, making it simple to construct fabricated visual images automatically [20]. DeepFakes are the common name for the videos produced with this technique, and there are numerous similar techniques. On social media applications such as Snapchat, the graphics-based face-swap approach [21] is a quick but subpar way that is employed. When faces are added to the learned frames from source videos or a collection of images, DeepFakes based on GANs [2] provide more desirable output. Both the FakeApp and the Faceswap Github host the method for public use [22]. In Face2Face [23], facial expressions are reenacted from source to target frames by an algorithm. These techniques allow for the merging of videos with separately fabricated sound data to produce fully fabricated material. Based on text or the words of another speaker, algorithms can produce speech that convincingly resembles a target speaker [24].

When detecting audio, interesting outcomes from the ASVSpoof challenges have been obtained [25], even if it is more concerned with stopping people from getting around voice biometrics. Although modern techniques attain very high accuracy, they do not provide a comprehensive solution that is resistant to various video modifications, and as a result, they are not trustworthy for spotting DeepFakes in the field.

According to the authors of [26], DeepFake detection was considered a binary classification problem and evaluated the ability of detection approaches to distinguish original videos from DeepFake videos, this technique was centered on the quality of image measures with an SVM classifier capable of detecting high-quality DeepFake videos with an equal error rate of 8.97%. However, this method's limitation comes with subjective evaluations to study the vulnerability of human subjects to DeepFakes, which are needed, and the method needs a more robust detection algorithm.

The authors of [27] developed a LAIR dataset which was applied to refine data fed into the language model to generate arrays of words, which are then fed into the deep learning models. This method needs an increase in detection accuracy by merging the gated recurrent unit (GRU) and CNNs to obtain the best result.

A rather simple but efficient method was created by the authors of [28]. This combines CNNs, RNNs, and, DFDC Dataset to give the best possible result as at this time. The system employs a single GPU to process videos rapidly (in less than eight seconds), but it only concentrates on face modification detection and ignores any audio content analysis, which might enhance detection accuracy significantly in the future study. The study was

proposed to enable national and international journalists to explore the requirements to craft a device for the spotting of DeepFake videos. The authors in [29] studied the directions of data-oriented, feature-oriented, model-oriented, and application-oriented. The recall, or the proportion of tagged fake articles that are expected to be misleading, was used to gauge the study's sensitivity. The authors of [30] conjoined findings based on previous studies and outcomes to establish a scheme and functionality layout for the detection platform, but this method has no clear cut methodology to follow or the approach to follow in building the said tool.

The authors of [31] analyzed numerous neural network-based strategies in the context of DeepFakes categorization in high compression instances and illustrated that a recommended metric learning-based strategy can be very competent in fulfilling such a characterization. The metric learning method using a triplet network architecture proved beneficial when fewer frames per video were used to assess its realism, but a significant constraint of the approach is its generalizability across different datasets. It lacks an unsupervised feature modification to acclimate the feature space from the source dataset to the target dataset, which would make the model more robust and label self-sufficient.

According to the authors of [32], experiments revealed that social framework and distribution are crucial elements allowing highly accurate discovery of news at an early stage, after only a few hours of transmission (92.7% ROC AUC), and detection of misinformation but the study did not explore further beyond fake news detection. The model's applications in social network data processing include journalistic subject categorization and virility prognostication.

However, the authors of [22] obtained temporal data; a time-distributed FaceNet previously trained on VGGFace2 was followed by a unidirectional LSTM layer. They constructed their own Facenet LSTM model, which aims at temporal inconsistencies in the videos but did not check whether the tool can detect new variations of fake videos in addition to the ones it has been trained on. A model was developed by the authors of [5] which was based on artificial intelligence and error level analysis (ELA) detection; it is related to entropy and information theory, such as the cross-entropy loss function in the final softmax layer, constrained mutual information in data preprocessing, and some information theory-based encoder applications. This model must be evaluated to improve the overall effectiveness of video compression detection methods.

The authors of [4] used an X2Face method and a first-order motion model for image animation) and achieved DeepFake video deconstruction using a SIFT (scale invariant feature transform)-based technique The limitation is aimed at obtaining a more comprehensive dataset for testing from both historical and animated personalities, as well as an explication of other potential techniques and methodologies for DeepFake video detection.

A model was designed and implemented by the authors of [33]: a deep-fake identification technique with mouth features (DFT-MF), employing a machine learning approach for detecting DeepFake videos through reclusion, analysis, and confirmation of lip/mouth motion. However, this method only focuses on just the mouth and not the entire face and body movements.

The authors of [34] looked into intelligent amplified CNN state-of-the-art innovation used for real-time reconstruction of DeepFake visual elements utilizing components such as video cameras and surveillance cameras. The study mixed easily with an augmented DeepFake arrangement with partition while expanding accuracy to 95.77% but the contrast in the calculated expenditure of the actualized technique is low. The study was proposed to solve facial forgery identification by determining if a photograph or video has been tampered with by DeepFake; this uses the ensemble technique for identifying DeepFake videos or images using image partitioning and separable convolutional neural network (CNN) [35]. While it proved productive, it was suggested that future works could be tailored towards extending image forgery detection to parts of a human body besides the face, along with trying to improve the prediction performance of the classifier.

The authors of [36] proposed using adversarial apprehension to improve DeepFake images and trick common DeepFake identifiers. This method uses the fast gradient sign method (FGSM) and the Carlini and Wagner L2 norm attack in both its Blackbox and Whitebox settings, while this method's limitation came about as a result of the excessive overhead required to process a single image and further experimentations are needed to project the achievement on adversarial attacks into other domains.

The authors of [37] produced a study that focuses on identifying the existing DeepFake detection framework's limitations and shortcomings. The theoretical and empirical analysis of their ideal conventional systems and datasets reveals that the use of Face-Cutout can contribute to the overall data variation and mitigate the problem of clustering while attaining a decrement in LogLoss of 15.2 percent to 35.3 percent on various datasets, the future aim will be to delve into the use of this augmentation principle on more DeepFake datasets. The authors of [38] proposed a DeepFake identification strategy that makes use of a 3D-attentional inception network, this technique encapsulates both temporal and spatial information concurrently with the 3D kernels to improve detection capabilities. The conducted experiments demonstrated that the method obtains parallel cross-functional dataset performance using cutting-edge techniques and, in the future, the model can be geared towards other aspects that use DeepFake such as audio and texts.

Similarly, the authors of [7] proposed a convolutional vision transformer for detecting and identifying DeepFakes which is a generalization technique for DeepFake video detection using CNNs, convolutional neural network (CNN) and Vision Transformer are the two components of the Convolutional Vision Transformer (ViT). The CNN obtains trainable characteristics, whereas the ViT takes the acquired parameters and summarizes them with the help of an attention mechanism but the model can be improved by newer datasets developed in the nearest future.

In [15], the authors suggested a temporal-aware network for automatically detecting DeepFake videos. This system extracts attributes at the frame level using CNN. These characteristics are learned through recurrent neural networks (RNNs), which learn to recognize whether or not a video has already been tampered with. This system can produce a competitive result while using a simple channel architecture. The future direction of this project is to examine how to improve the robustness of the system against DeepFake videos using unseen techniques during training. The authors of [39] proposed that to aggregate multi-frame features to detect DeepFake videos, they solved this problem from the set outlook and proposed set convolutional neural network, a new design based on a set (SCNN). As an example, the approach they offered is an amalgamation advancement of a single-frame digital video analytics network and, meanwhile, it was noticed that a better forbearance network can produce much-desired results. As a result, the search for improved network infrastructure will be the next step in the process.

Another perspective was considered by [40] when they presented a study applicable to convolutional neural networks (CNNs) using transfer learning, which involves initializing pre-learned weights for upper layers of deep CNNs. This gave a better outcome in shorter training durations than CNN-trained models on nonlinear mapping weights for the detection of DeepFake films. However, it is still believed that the model's fitness can be enhanced by using ConvLstm2D (Tensorflow) layers and providing image sequences to the network instead of a single image, which would address temporal incompatibilities in DeepFake movies and (LSTM) along with characteristic distortion. The authors of [41] proposed distinguishing authentic movies or images from DeepFakes, presenting a new Patch&Pair convolutional neural network (PPCNN). Instead of learning the entire face in this method, it isolates the face into smaller frames before sending the face repairs to the network. However, the PPCNN is efficient when detecting DeepFake videos originating from the same dataset and the generalization of the model can be improved on a two-branch learning framework on cross-origin DeepFake videos.

Another perspective was described by [42], which is an artificial intelligence method for identifying artificial intelligence-generated DeepFake videos from real videos. The

technique displayed the advertency that the DeepFake algorithm at the time could only produce images having limited resolutions, in which further warping was needed to reflect the primary video's genuine faces. Since it is using a predesigned network scheme for this task (e.g., resnet or VGG), there is a need to appraise and ameliorate the robustness of the detection method with several levels of video compression. The authors of [43] pointed out a developing challenge of partisan face manipulation in DeepFake videos, which simply gives video-level descriptions and does not manipulate all of the faces in the fake videos. The study addressed the DeepFake problems by combining instances of faces and input videos treated as bags and instances in this learning framework. In contrast to classic MIL, which produces a direct mapping from instance integration to instance projection and then to bag prediction, a sharp MIL (S-MIL) is presented, which builds a direct mapping from instance embedding to bag prediction but the constructed FFPMS dataset used is yet to be subjected to rigorous testing by various platforms and DeepFake detection techniques.

To aggrandize the effectiveness of identifying DeepFake-created face-barter images from actual ones, a novel imitation attribute extraction technique based on deep learning and error level analysis (ELA) was developed by the authors of [44]. The ELA different image encoding ratios can be detected using this method. The CNN extracts the fake features and determines the genuineness of the images. Though the method proposed is appropriate under lossy compression, image tampering detection is possible, but not ideal under poor quality or lossless encoding. The authors of [45] proposed a graph-based model which uses the factual scheme of a textual document for DeepFake detection. Further analysis of the model showed that this model can tell the difference between computer-generated text and human-written content in terms of factual structure but the model can be rebuilt on a more robust dataset and subjected to rigorous tests on a different source of electronic printed texts. The authors of [46] introduced a study that demonstrates how the *Wall Street Journal*, *Washington Post*, and Reuters, together with three of the largest Internet-based firms, Google, Facebook, and Twitter, are coping with the rise of DeepFakes as a new form of fake news. The study focuses on the techniques for detecting DeepFakes, as well as the ramifications of DeepFakes on democracy and national security. However, both the digital platforms study and the media samples revealed a western-oriented cultural drift. This may make it difficult to extrapolate results to similar organizations in other countries.

The authors of [47] review various works in DeepFake films and photographs while analyzing how they are made. Furthermore, the study examines the impact of Deep-Fake on societal structure in terms of security and its application. The study was able to thoroughly dissect the meaning of DeepFake, the application areas, and also the misuse of the technology in a comparative manner but this study can be tailored towards the new methods being used to detect DeepFakes and the methodologies employed can be evaluated comparatively.

Multiple graph learning neural networks (MGLNNs) are a revolutionary learning framework put forth by the authors of [48], to classify data using various views of a graph. The objective of a MGLNN is to simultaneously combine multiple graph learning and various graph structures to learn an ideal graph structure that best suits GNNs' learning. The suggested MGLNN is a generic framework that may be used to handle multiple graphs using any particular GNN model. The suggested MGLNN model has also been trained and optimized using a general approach. Results from experiments on various datasets show that MGLNN performs better on semi-supervised classification tasks than some other related approaches.

The authors of [49] have suggested the use of the precise real-time object detection framework WilDect-YOLO for finding many classes of endangered wildlife species. The integration of DenseNet, spatial pyramid pooling, and redesigned path aggregation network has improved the performance of the entire network. The suggested model outperforms current state-of-the-art models with mAP and F1-score of 96.89% and 97.87%, respectively, at a detection rate of 59.2 FPS.



The precise single stage detector is a novel design presented by the authors of [50], which is a modified version of the single shot multibox detector (SSD) (PSSD). Real-time performance of the suggested model, PSSD, is impressive. In particular, PSSD outperforms cutting-edge object detection models by achieving 33.8 mAP at 45 FPS speed on the MS COCO benchmark and 81.28 mAP at 66 FPS performance on Pascal VOC 2007 using Titan Xp hardware and 320 pix input size. The suggested approach also performs noticeably well with bigger input sizes. PSSD can achieve 37.2 mAP with 27 FPS under 512 pixels on MS COCO and 82.82 mAP with 40 FPS on Pascal VOC 2007. The outcomes of the experiment demonstrate that the suggested model offers a better balance between speed and accuracy. Table 1 gives a summary of the related work in the literature.

**Table 1.** Review of selected related work in DeepFake Detection.

| Author(s) | Technique(s) | Strength | Gap |
|---|---|---|---|
| Đorđević et al. (2020) [4] | X2Face method, first-order motion model, SIFT (scale invariant feature transform). | It employs rotation invariant screening, which has the potential to distinguish original from DeepFake films. | Future work can be geared towards obtaining a more comprehensive dataset for evaluating historical and fictional personalities, as well as a study of other possible traits and approaches for detecting DeepFake videos. |
| Zhang et al. (2019) [5] | CNN and error level analysis (ELA) | When compared to the most up-to-date models, this technique has few layers, the training time is shorter, and has higher efficiency. | The reliability of video compression-detecting methods needs to be evaluated and improved. |
| Sohrawardi et al. (2020) [6] | Qualitative interviews supplemented engaging concepts or early versions of the standard option tools. | It built the foundation on which DeepFake detection tools can be built. | No clear cut on the methodology to follow or the approach to follow in building the said tool. |
| Wodajo et al. (2021) [7] | Convolutional Neural Network (CNN) and Vision Transformer (ViT). | On the DFDC dataset, the authors introduced a CNN layer to the ViT framework and produced a reasonable result. | This method could be used to build on previous work by incorporating more datasets supplied under the DeepFake study, making it more broad, precise, and resilient. |
| Güera et al. (2018) [15] | CNN and RNN | The system achieves a competitive result while using a simple pipeline architecture. | The proposed system can explore how to improve the efficiency of the system against counterfeiting videos using unseen techniques during training. |
| Korshunov et al. (2018) [26] | Pre-trained generative adversarial network (GAN) | The technique is with an 8.97 percent identical error rate, and picture quality metrics using an SVM classifier can recognize HQ DeepFake films. | In the future, new generic methods and databases need to be developed. Subjective evaluations of human subjects need to be carried out to study the vulnerability to DeepFakes. |
| Girgis et al. (2018) [27] | RNN (recurrent neural network), LSTM (long short-term memories), Vanilla, GRU (gated recurrent unit), CNN (convolutional neural networks) | LAIR dataset is being applied to prep data to word embedding to obtain word vectors, which are subsequently fed into the deep learning algorithm. | In the future, the system can be aimed to increase the detection to get the best results and combine GRU and CNN's accuracy. |
| Montserrat et al. (2020) [28] | MTCNN, CNN, AFW (automatic face weighting), and GRU (gated recurrent unit). | This method combines CNNs, RNNs, and DFDC Dataset to give its results, it also uses a single GPU to process videos quickly (in less than eight seconds). | This study concentrates on facial modification recognition and ignores any audio content analysis, which could enhance detection capability significantly in the future. |
| Shu et al. (2017) [29] | Characterization and detection. | The sensitivity of this study was determined by the recall, which is the percentage of marked fake articles that are anticipated to be misleading. | The study directions are data-oriented, feature-oriented, model-oriented, and application-oriented. |

**Table 1.** *Cont.*

| Author(s) | Technique(s) | Strength | Gap |
|---|---|---|---|
| Kumar et al. (2020) [31] | Metric learning a network architecture based on triplets | This approach is useful for social data compression and is usually unavoidable on media platforms | It can use an unsupervised method to make the design resilient and label independent, domain adaptation is used to construct the feature space from the source dataset to the target dataset. |
| Monti et al. (2019) [32] | Geometric deep learning | Because it is built mostly on connectedness and distributing traits, this approach has the potential to be language and geographic-agnostic. | This study intends to look into other uses for the model in social network data analysis, such as identification and features and capabilities prediction, beyond just detecting bogus news. |
| Sohrawardi et al. (2019) [22] | Time-distributed FaceNet Pre-trained on VGGFace2, and unidirectional LSTM. | Results displayed on both within and discordant datasets, near to an accurate detection was achieved. | It should be established that the features can recognize new forms of false movies in addition to those on which they have been trained. |
| Elhassan et al. (2020) [33] | CNN and DFT-MF model | This study demonstrates an improvement in some other approaches that were used, and the results were compared to demonstrate the performance improvements. | The experiment of the study should focus more on the entire facial movements and not just the mouth. |
| Ahmed et al. (2021) [34] | Rationale augmented convolutional neural network (CNN) on MATLAB R2019a platform | This study supports a progressively DeepFake layout was completed, with a division enlarged accuracy of 95.77 percent. | The difference in calculation costs between the two actualized strategies is small; the CNN technique is almost complete. |
| Yu et al. (2019) [35] | Ensemble model, separable convolutional neural network (CNN), and image segmentation. | The ensemble model improves detection capabilities and the study results indicate that the proposed solution performs well. | Future studies will focus on expanding picture forgery detection to sections of the human body besides the face, as well as increasing the generalization capacity of the trained model. |
| Gandhi et al. (2020) [36] | Carlini and Wagner L2 norm attack in the BlackBox and Whitebox settings, and Fast Gradient Sign Method. | This study explores two advancements to DeepFake detectors: The first is Lipschitz regularization, and (the other is deep image prior (DIP). | In other domains, further experiments would be needed to show the success of adversarial attacks. Another limitation is time consumption when processing a single image. |
| Das et al. (2021) [37] | quantitative and qualitative analysis | When compared to existing occlusion-based algorithms, our method found a significant reduction in LogLoss of 15.2 percent to 35.3 percent on various datasets. | In the future, the study can be geared towards examining the application of this enhanced policy to a wider range of face modification and counterfeit datasets |
| Lu et al. (2021) [38] | With the 3D kernels, you may get temporal and spatial information in real time. | This study improves the detection capability. Experiments demonstrated that the method eloquently on a cross-dataset evaluation outperforms the competition. | In the future, the model can be diverted to be used in other aspects that use DeepFake such as audio and texts. |
| Xu et al. (2021) [39] | Set convolutional neural network (SCNN). | The suggested technique is a single-frame digital video analytics system fusion promotion. | It was noticed that a single-frame digital video analytics system fusion promotion approach is proposed. This will greatly improve this framework. |
| Suratkar et al. (2020) [40] | Convolutional Neural Networks (CNNs) with transfer learning. | Initializing weights pre-trained Deep CNNs with shallow layers produce better outcomes in shorter training times than CNN models with nonlinear mapping weights. | The system's robustness can be increased by using ConvLstm2D (Tensorflow) layers and sending a sequence of photos to the network rather than a single image. |

**Table 1.** *Cont.*

| Author(s) | Technique(s) | Strength | Gap |
|---|---|---|---|
| El Rai et al. (2020) [41] | CNN with transfer learning. | The results obtained showed relevant accuracy when compared with other competitive methods. | This method can be used to analyze all of the movies in the datasets. |
| Li et al. (2018) [42] | Resnet or VGG and CNN | This strategy keeps costs down and time while collecting training data. | There is a need to examine and adjust the detection robustness method for multiple video compression. And it would need to explore a dedicated network structure for the detection of DeepFake videos. |
| Li, et al. (2020) [43] | Multiple instance learning framework (MIL) and Sharp MIL (S-MIL). | S-MIL overcomes the limitations of performance on single-frame datasets when used in classic DeepFake image recognition applications. | The constructed FFPMS dataset was not exposed to rigorous testing by different platforms and DeepFake detection techniques. |
| Zhang et al. (2020) [44] | CNN, and Error Level Analysis (ELA). | When compared to conventional method detection methods, the strategy effectively extracts the counterfeit attribute and hence outperforms them in terms of simplicity and efficiency. | The presented method used in this study is effective in detecting picture alteration when lossy compression is used, however, it is not optimal for detecting tampering when lossless or low-quality compression is used. |
| Zhong et al. (2020) [45] | Graph neural network. | Concept analysis also reveals that the system can differentiate between artificially intelligent content and human-written text in terms of factual structure. | In the future, the model can be built on a more robust dataset and tested on several sources of electronic printed texts. |
| Vizoso et al. (2021) [46] | Extensive qualitative analysis | This study applies a comprehensive method to examine how significant communication firms are attempting to combat the dissemination of DeepFakes, the most recent and technologically based kind of misinformation. | Both the media and internet platforms analyses revealed a foreign cultural bias in the population samples. This could significantly affect the findings of similar groups in other countries. |
| Albahar et al. (2019) [47] | Google's Image Lookup had been used to look for source material on various social networking sites, and then it replaced data of faces on its own. Because the algorithm is built on machine learning, it has a high success rate. | This study has been able to dissect the meaning of DeepFake, the application areas, and also the downside or misuse of the technology in a comparative manner. | Future applications of this study can be geared towards the new methods being used to detect DeepFakes and a comparative evaluation of the methodologies. |
| Li et al. (2020) [51] | Patch&Pair Convolutional Neural Network (PPCNN) | When it comes to describing DeepFake videos from the same-origin dataset, PPCNN is productive, and the multiple training structure can help the model generalize to cross-origin DeepFake videos. | In the future, by combining forensic technologies, it will develop more general models to detect complex DeepFakes with different compression levels and resolutions. |

## 3. Materials and Methods

Convolutional neural networks have been exceptionally successful in image analysis. The term refers to a specific network architecture that has a class in neural networks, the first stage of each so-called hidden layer is the local convolution result of the recent layer (the kernel includes trainable weights), and the second phase is the max-pooling stage, which decreases the number of subunits by maintaining just the maximum response of many units from the first stage. After multiple concealed layers, the last layer is comprised of a completely linked layer. It consists of a unit for each category that the system detects, and each of these units receives input from all preceding layer units.

It is commonly known that intricate DL-based architectures with many hidden layers, such as Alexnet and VGG16, can effectively handle a high number of classes. However, when there are fewer classes, they have a tendency to overfit, which reduces accuracy. The high storage needs are also a result of the huge number of layers. In this study, a lightweight DL-based architecture has been suggested due to the dataset's low number of DeepFake classes.

### 3.1. The Proposed Convolutional Neural Network

For DeepFake video identification, several CNN models with a limited number of layers were used. Several versions with various numbers of filters for each layer were taken into consideration, even though the number of layers selected was five. The filter size was maintained at $3 \times 3$ in each layer, with a $2 \times 2$ max-pooling layer coming after each convolution layer. The data were flattened in 2D after being processed using convolution and maximum pooling. Data were then sent to a dense layer with 128 nodes after flattening. Figure 2 displays the CNN architecture.

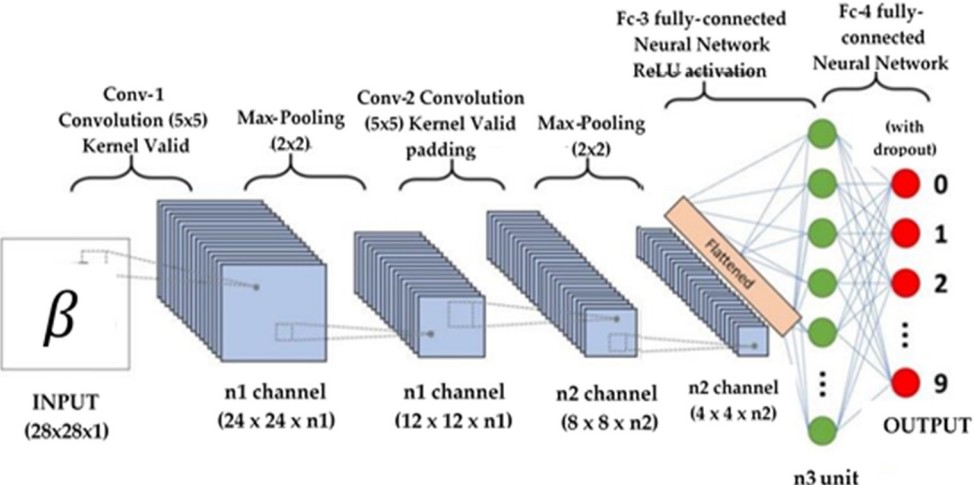

**Figure 2.** Convolutional neural network architecture.

In a neural network, the CNN is a subset of artificial learning networks that is most typically employed to assess visual imagery. Matrix computation can be compared to neural networks, but this is not the circumstance with ConvNet. The special technique used in this is convolution.

Convolutional neural networks are a collection of artificial neuron layers that function together. Artificial neurons are mathematical functions that examine the weighted total of aggregate inputs and then output an activation value, similar to their biological counterparts. When an image is fed into a ConvNet, each layer generates several activation functions, which are then transferred onto the next layer.

Essential features such as horizontal or diagonal edges are extracted in the first layer. This information is passed on to the next layer, which is responsible for detecting more complicated features such as edges and combinational edges.

The layer categorization provides a series of confidence ratings (numbers between 0 and 1) relying on the activation map of the previous convolution layer, indicating how probable the image is to conform to a "class." A nice example is a ConvNet that recognizes cats, dogs, and horses, with the last layer's output being the possibility that the input image features any of those species. Figure 3 shows the classification layers combination for the proposed model.

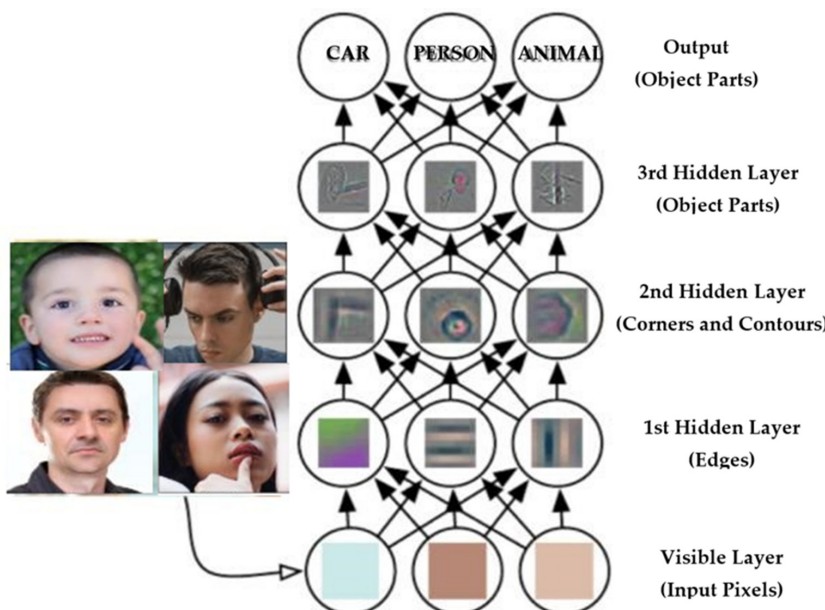

**Figure 3.** Classification layer combination.

　　As with the convolutional layer, the pooling layer makes sure that the spatial size of the convolved feature is minimized. The amount of computing power needed to process the data is decreased by reducing its size. The two types of pooling are average and maximum. Max pooling is used to obtain the highest value of a pixel from a portion of the image that is represented by the core. Max pooling also functions as a noise suppressant. It eliminates noisy activations while simultaneously performing de-noising and complexity reduction.

　　The average of all the values from the area of the image covered by the kernel is what is returned by average pooling, on the other hand. Dimensionality reduction is all that average pooling does to reduce noise. Therefore, we can conclude that max pooling outperforms average pooling significantly. Despite their strength and resource sophistication, CNNs deliver in-depth findings [52]. It all comes down to identifying patterns and traits that are minute and insignificant enough for the human eye to miss. However, it falls short when it comes to understanding the substance of digital photographs. The maximum and average pooling used by CNNs is depicted in Figure 4.

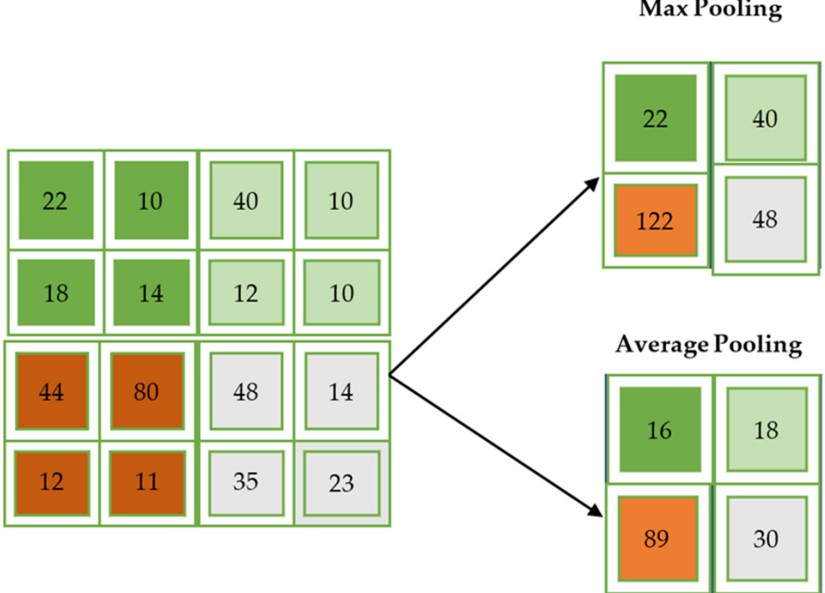

**Figure 4.** Maximum and average pooling.

These limitations are clear when it comes to practical application. For example, social media content was frequently filtered using CNNs. They were still unable to completely prevent and erase inappropriate material, despite having been trained on a significant number of images and videos. For example, a 30,000-year-old sculpture had been labeled as nudity on Facebook.

### 3.2. The Proposed CNN Enhanced with ReLU Architecture

The model is based on a well-performing image classification network that switches the convolutional layers and pooling layer for extraction features and a classification network [53]. This network will start with a sequence of five successive convolution layers, batch normalization layer, and pooling layer, because existing image analytics methodology easily diminishes its capacity to detect DeepFake in movies due to compression, which often degrades the data. One hidden layer was used to construct a dense network.

For better generality, ReLU activation functions are added to the convolutional layers to generate a non-linear, batch normalization layer and pooling layer. Robustness is improved by fully-connected layers using a neural network regularization technique by discarding a random subset of its units; this technique is called dropout.

Consider the neural network's convolutional layer. Each of these layers consists of the input image convolutioned with a series of convolutional filters, biases added, and a nonlinear activation function applied. The incoming image could, for instance, be multichannel and colored. One convolution filter's use can be explained as follows:

$$O(x, y) = \sum_c \sum_{\Delta x} \sum_{\Delta y} I(c, \ x + \Delta x, \ y + \Delta y) w(c, \ \Delta x, \ \Delta y), \tag{1}$$

$O$ is the convolution result, $c$ is the channel number, $I$ is the input image, and $w$ is the filter matrix, where $(x, \ y)$ is a point on the output image. Due to the fact that it has different coefficients for the various input image channels, the filter $w$ itself can also be regarded as multichannel. The output of the convolution is then added to, and a nonlinear activation function $\varphi$ is applied to:

$$O'(x, \ y) = \varphi(O(x, \ y) + b) \tag{2}$$

where $O'$ are the output values of the convolutional layer for the first filter; $b$ is the bias vector; and $\varphi$ is the activation function, for example, ReLU or a hyperbolic tangent. The output of the convolutional layer can also be thought of as multichannel since the neuron network often contains several filters, one channel from each filter. Let us analyze a convolutional layer's calculation complexity. Assume that the input image is $(N \times M)$ pixels in size, the filter size is $(K \times \text{K})$, there are C channels, and there are L filters. The primary complexity of the layer will then consist of $O(NMLK^2C)$.

The CNN model of a network-in-network (NIN) with ReLU is used to examine the input distributions on non-linear activation layers. In this design, batch normalization and dropout are added. Convolutional, batch normalization, and non-linear activation layers make up each "Conv" layer, save the last one (i.e., ReLU). The convolutional layer alone makes for the final "Conv" layer (conv5-5). The following is a definition of ReLU:

$$y_i = \begin{cases} x_i, \ if \ x_i > 0, \\ 0, \ if \ x_i \ \leq 0, \end{cases} \tag{3}$$

where the $i$th channel's ReLU's input and output, respectively, are $x_i$ and $y_i$. ReLU's output is active and equal to the input if the input is greater than 0. ReLU's output is deactivated and equal to zero if its input value is less than 0. So, the hard threshold zero is what makes

ReLU nonlinear. Assuming that the basic input $x_i^0$ and the jitter (or noise) $n_i$ make up the input $x_i$ of ReLU. Thus, $x_i = x_i^0 + n_i$ ReLU can then be rewritten as follows:

$$y_i = \begin{cases} x_i^0 + n_i, \ if \ x_i^0 + n_i > 0, \\ \quad 0, \ if \ x_i^0 + n_i \ \leq 0, \end{cases} \tag{4}$$

In actuality, the jitter or noise $n_i$ is negligible. The jitter (or noise) $n_i$ may result in being mistakenly triggered or deactivated when $x_i^0$ is close to zero. For example, when $x_i^0 = 0.5$, it should be activated. The small jitter $n_i$ will, however, unintentionally not be engaged if it is less than $-0.5$. Similarly to that, it should not be activated when $x_i^0 = -0.5$. The small jitter $n_i$ will, however, unintentionally activate if it is more than 0.5.

Since the majority of the reported inputs to ReLU are concentrated close to zero, hence, the majority of ReLU outputs are sensitive to a tiny jitter. Therefore, the learned CNN with ReLU is probably susceptible to jitter or noise.

The essential need for a dropout and maximum pooling layer in this technique is to ensure that unnecessary random subsets are eliminated from the unit and the algorithm can focus on only the important aspect to generate an optimal solution, this normally would generate a major concern due to the elimination of some data but this has been taken care of by the convolutional layer which is only focusing on the part that matters in the detection, that is, the face. The convolution blocks will have the size and the number of filters to be used in the convolution, these filters will identify the existence and location of image features present in the video frames. By standardizing the inputs to each layer of the network, the batch normalization layer increases the speed, performance, and stability of the neural network, reducing the interdependence of the parameter of one layer in the input distribution of the next layer. The internal covariant shift is the term for this dependency, which has a disruptive influence on the learning process.

To address this compression issue, in the output layer, the activation function is sigmoid, the convolution layer employs leaky ReLU, and the loss function is mean square error. This study also utilizes extensive secondary data to fill the dataset, these data are videos produced from several DeepFake algorithms and other videos which are not altered. Figure 5 shows the architectural structure of the proposed system.

### 3.3. The Dataset Description

This study employed three datasets to train and test the method on various objects. The system shows a high capability of delineating videos having higher resolution compared to the experiments.

DeepFake Dataset

This dataset was created by the authors in [9], and it was used for developing their DeepFake detection system called MesoNet. This was accomplished by teaching autoencoders to perform the task; for a realistic result, several days of practice using processors were required, and it could only be accomplished for two faces at a time. The study chose to download video profusions available to the public online, to have enough variety of faces. Therefore, the study collected 175 forged videos across different platforms.

The video's minimum standard resolution is $854 \times 480$ pixels, and its lengths range from two seconds to three minutes. All the videos were compressed in different compression levels using H.264 codec. A trained neural network for facial landmark detection was used to organize the faces after they had been extracted using a Viola–Jones detector. On average, about 50 faces were extracted from each scenario. In conclusion, this dataset was reviewed manually to eliminate misalignment and wrong face detection, while to avoid having the same distribution of image resolutions either good or poor, both classes were used to avoid bias in the classification task.

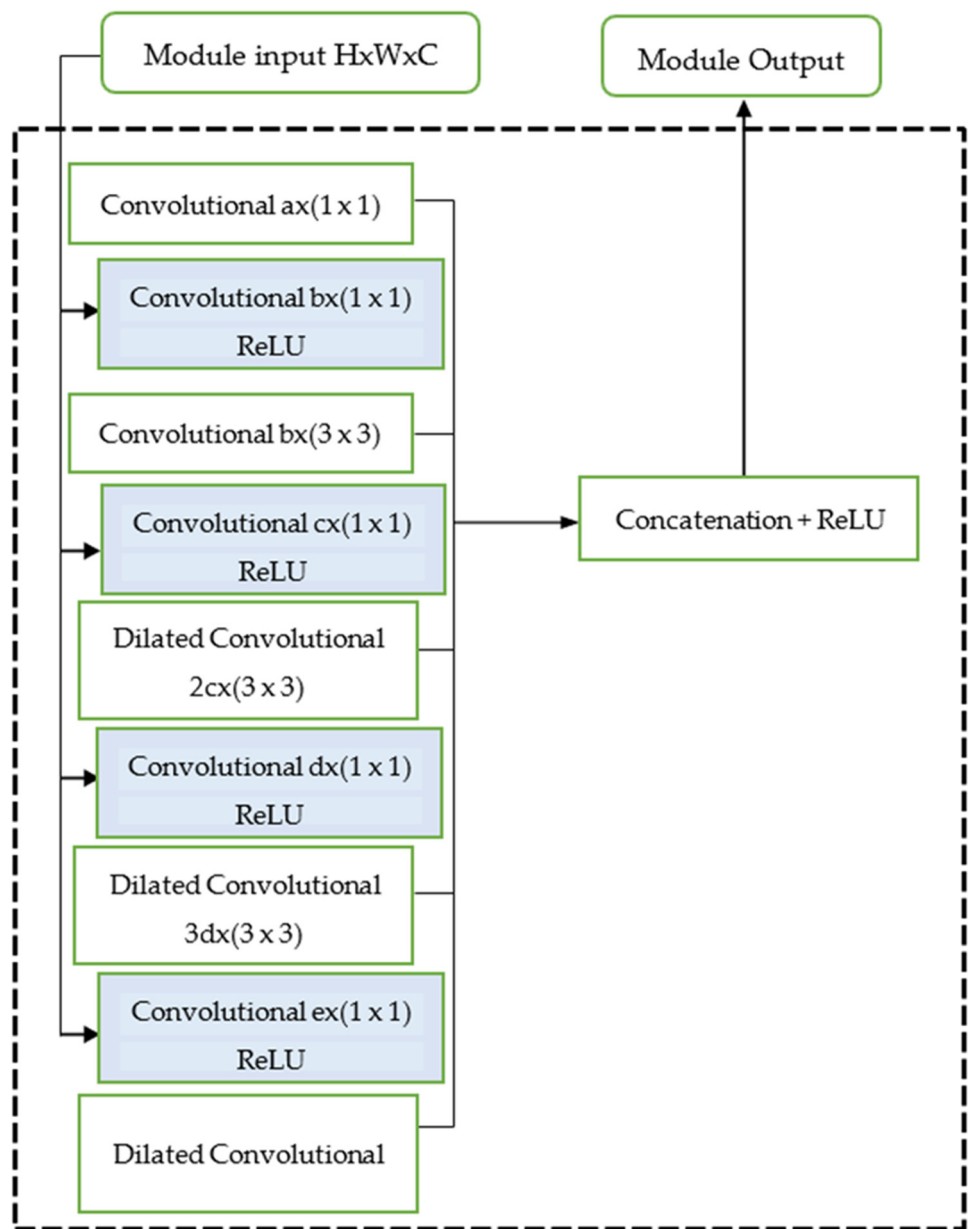

**Figure 5.** Proposed design architecture.

Face2Face dataset

To ensure that the proposed method would be able to detect other face forgeries, the Face2Face dataset has several hundred ranging to thousands of forged videos. This dataset has already been divided into a testing, training, and validation set. One major advantage of the Face2Face set is that it provides lossless, already compressed videos; this enables the study to test the system's robustness with different levels of compression. In the training, only about 300 videos were used. The model was assessed using the 150 fabricated videos and their originals from the testing set.

First-Order Motion Dataset

The first-order motion model was trained and set up on four different datasets that are the VoxCeleb dataset, which is a dataset for faces and has 22,496 videos. These videos were extracted from YouTube. The third dataset included was the BAIR robot pushing dataset, which features films taken by a Sawyer robotic arm pushing various things across a table. The UvA-Nemo dataset is a facial analysis dataset with 1240 videos. It includes 42,880 training videos and 128 exam films. Another collection of 280 tai-chi films was

gathered from YouTube, with 252 being utilized for training and 28 being used for testing. The first-order motion was used to create several DeepFake videos to serve as data used for testing the method as external data outside the development environment.

### 3.4. Setup of Classification

In this proposed system, an image dimension of $256 \times 256$ and weight optimization using ADAM with default parameters were accomplished using three color channels, red, green, and blue ($\beta = 0.9 \ and \ \beta_2 = 0.999$). The Keras 3.7.2 module was used to implement the system in Python 3.9. It uses the learning rate of 0.001 and uses mean square error to calculate the loss.

$$L = \frac{1}{N} + \sum_{i=1}^{N} y_i.log(pi) + (1 - y_i) \, . \, log(1 - (pi)) \tag{5}$$

As a result, the goal for $N$ movies in an input batch is to reduce data loss as much as feasible, where $pi$ is the label and $x$ is the prediction of the $i$-th video. The predictive algorithm and trainable parameters, respectively, are F and W. In Table 2, it is worth noting that for both datasets, 15% of the training set was utilized during model validation tuning.

**Table 2.** The cardinality of each class in the studied datasets.

| Set | Real Class | Forged Class |
| --- | --- | --- |
| DeepFake training | 7250 | 5111 |
| DeepFake testing | 4259 | 2889 |
| Face2Face training | 4500 | 4500 |
| Face2Face testing | 3000 | 3000 |

Assuming that $X$ is the input set and $Y$ is the output set in this study, the random variable pair $(X, Y)$ will take the values in $X \times Y$. Using f as the classifier's prediction function for values in $X$ to the action set A; with $1(a, y) = 1/2(a, y)^2$, the chosen classification job is to minimize the error $E(f) = E[f(X), Y]$.

### 3.5. Performance Evaluation

The proposed model was evaluated using various performance metrics such as accuracy, sensitivity, specificity, and error rate, which were calculated using the confusion matrix in the following Equations (6)–(11).

$$Recall = \frac{TN}{TP + FN} \tag{6}$$

$$Specificity = \frac{TN}{TN + FP} \tag{7}$$

$$Precision = \frac{TP}{TP + FP} \tag{8}$$

$$Error = \frac{FP + FN}{TP + TN + FN + FP} \tag{9}$$

$$Accuracy = \frac{TP + TN}{TP + TN + FN + FP} \tag{10}$$

$$F1 - Score = \frac{2 \times Precision \times Recall}{Precision + Recall} \tag{11}$$

where True Positive ($TP$) is the number of records classified as true positive, True Negatives ($TN$) is the number of records classified as true negative, False Positives ($FP$) is the number of records classified as false positive, and False Negatives ($FN$) is the number of records classified as a false negative. The values of $TP$, $TN$, $FN$, and $FP$ are obtained from the confusion matrix of the model.

## 4. Results and Discussion

To prepare the image data to accept external videos other than the one used in training the model, the study adopted a rescaling pixel value (between 1 and 255) to a range between 0 and 1. Therefore, we created a separate list for correctly classified and misclassified images using the following labels; correct real prediction, correct DeepFake prediction, misclassified real prediction and misclassified DeepFake prediction, as shown in the images below. A list is created to keep track of which video frame falls into which category. A for loop is also implemented to sort the classification into one of these four categories as shown in Figures 6–9.

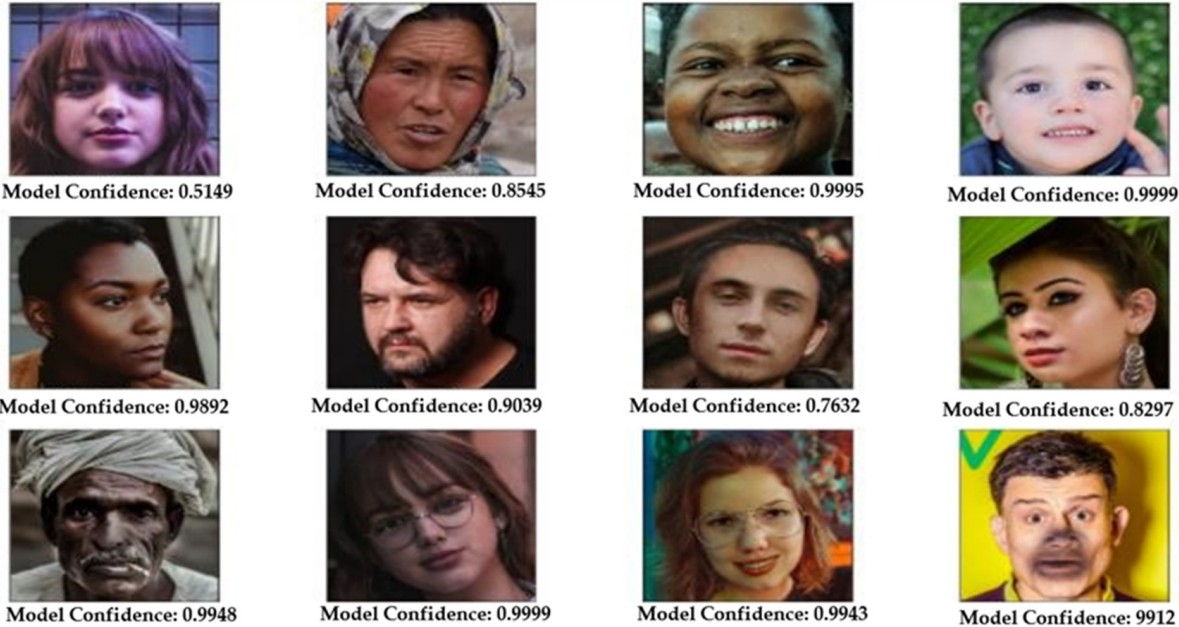

**Figure 6.** Correct real prediction: showing the model confidence in the prediction.

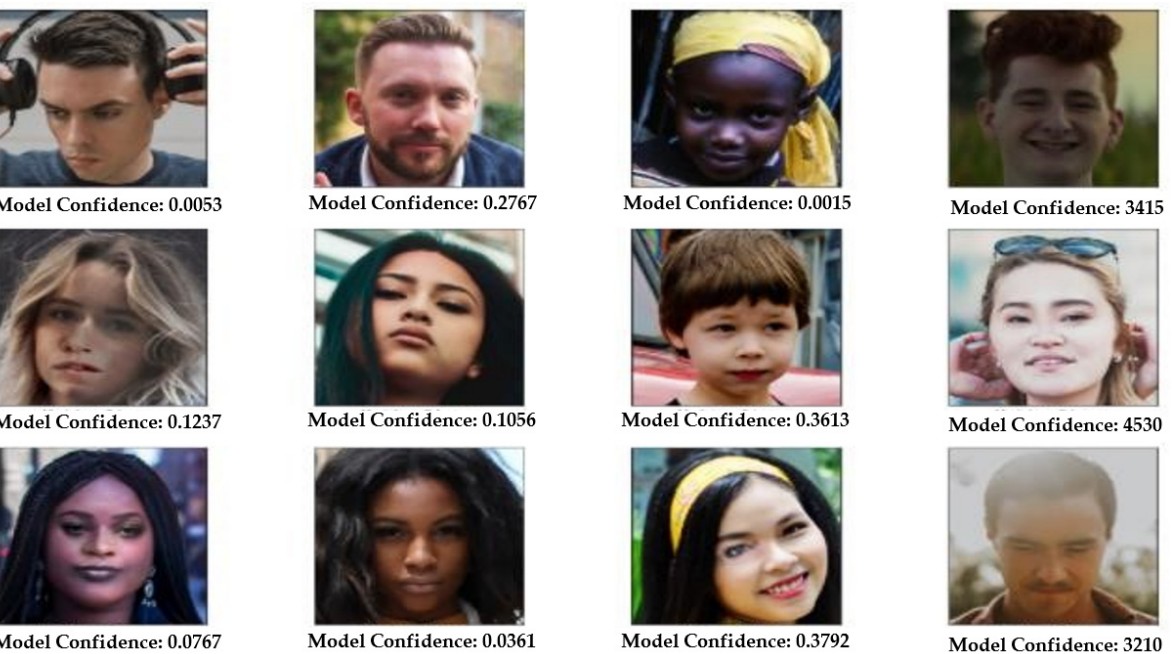

**Figure 7.** Correct DeepFake prediction: showing the model confidence in the prediction.

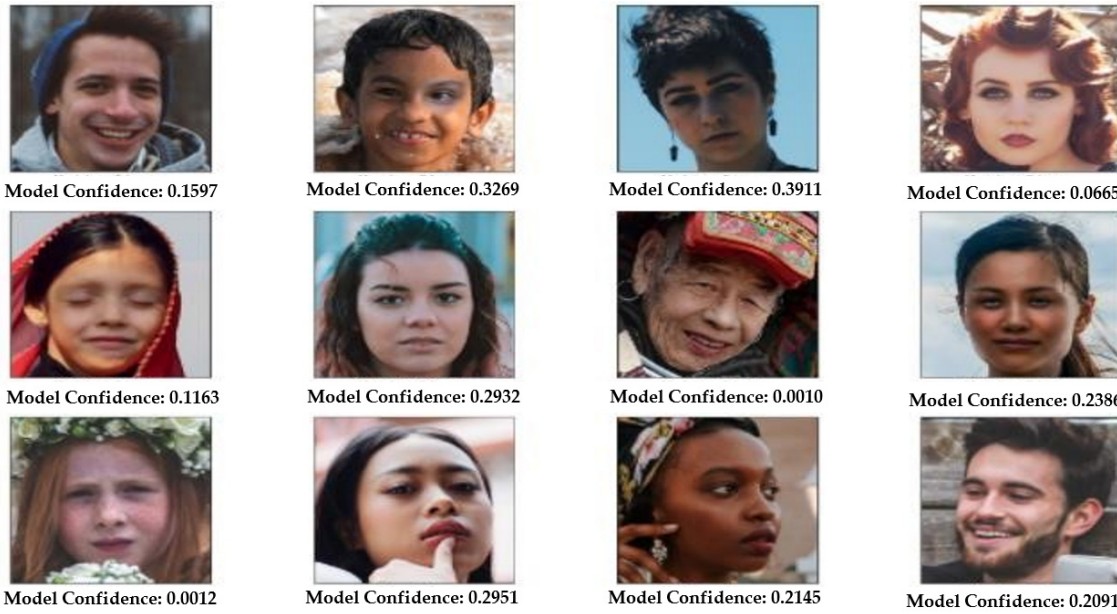

**Figure 8.** Misclassified real prediction: showing the model confidence in the prediction.

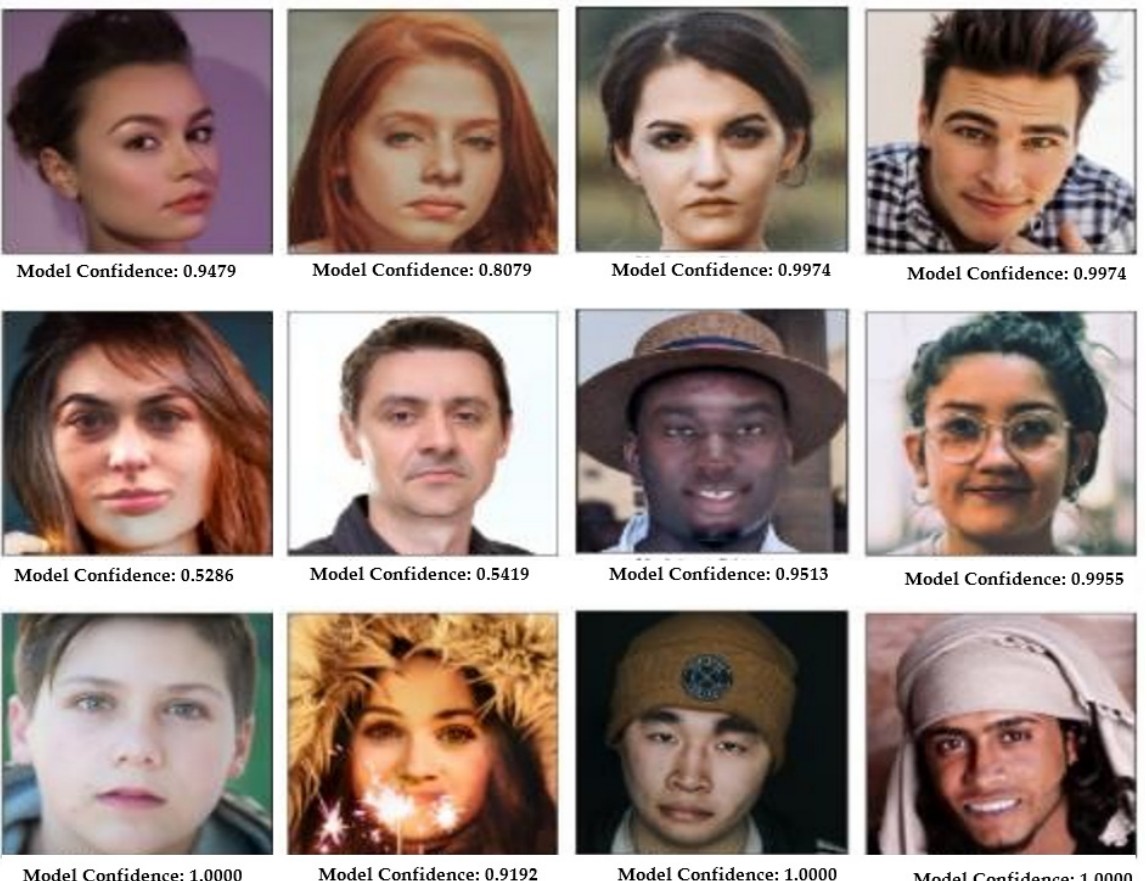

**Figure 9.** Misclassified DeepFake prediction: showing the model confidence in the prediction.

Compression, which results in significant information loss is a drawback of video analysis, particularly for online recordings. On the other hand, having a series of frames of the same face allows for the multiplication of perspectives and could lead to a more accurate evaluation of the film as a whole [9]. To accomplish this naturally, average the network prediction over the video. The frames of the same movie have a high correlation

with one another, thus therefore there is no rationale for a rise in scores or a confidence interval indication. In reality, the majority of filmed faces feature stable, clear frames for the comfort of the spectator [9]. Therefore, given a sample of video frames picked from the recording, the preponderance of accurate predictions can offset the effects of random misprediction, facial occlusion, and punctual movement blur. Results from the experiment are shown in Table 3. Both detection rates were greatly increased by the image combination. With the proposed network on the DeepFake dataset, it even reached a record-breaking 86%.

**Table 3.** Performance evaluation of our system.

| Accuracy | Precision | Recall | F1-Measure | Error | Specificity |
|----------|-----------|--------|------------|-------|-------------|
| 0.8632 | 0.8572 | 0.8768 | 0.8669 | 0.1368 | 0.8492 |

*Evaluation Results*

The classification results were obtained from a data split of 70% for training the dataset, 15% for validation, and the final 15% for evaluating the dataset. The DeepFake classifier receives the training set and uses it to create an optimum knowledge management pattern from the dataset. From the DeepFake dataset, the ROC is utilized to illustrate the classifier and pick the categorization threshold.

The study compares the result obtained from the experiment with those from other.

DeepFake detection models show the classification score for the trained network for Meso-4, MesoInception, and our system for the DeepFake dataset as shown in Table 3. When each frame is considered separately, the networks have attained a relatively identical score of around 90%. Due to the very low resolutions from some facial images extracted, a higher score is not expected. Table 4 illustrates the classification scores of multiple channels on the DeepFake dataset, taken separately for each frame. The table considers the Meso-4, MesoInception-4, and the proposed system.

**Table 4.** DeepFake Dataset classification scores.

| Model | DeepFake Classification Score | | |
|-------|------|------|-------|
| Class | Forged | Real | Total |
| Meso-4 | 0.882 | 0.901 | 0.891 |
| MesoInception-4 | 0.934 | 0.900 | 0.917 |
| Our System | 0.896 | 0.907 | 0.902 |

The findings for the three approaches of Face2Face forgery recognition are provided in Table 5. While the Meso-4 was able to obtain a 94.6% at the 0 levels of compression, MesoInception was able to obtain a 96.8% while our system achieved 98.6%. A visible degradation of scores is noticeable at the low video compression stage. However, the proposed method managed to fine-tune the classification and was able to obtain an 86.4% score at the compression level of 40. These comparisons are represented on the ROC curve in Figures 10 and 11.

**Table 5.** Face2face classification scores evaluation.

| Model | | Face2 | Face Classification S | Core |
|-------|---|-------|----------------------|------|
| | Compression level | 0 | 23 (low) | 40 (high) |
| Meso-4 | | 0.946 | 0.924 | 0.832 |
| MesoInception-4 | | 0.968 | 0.934 | 0.813 |
| Our System | | 0.983 | 0.920 | 0.864 |

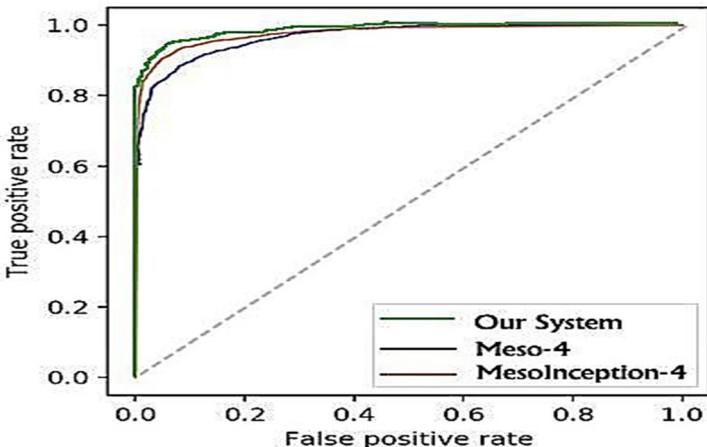

**Figure 10.** ROC curves of the evaluated classifiers on the DeepFake dataset.

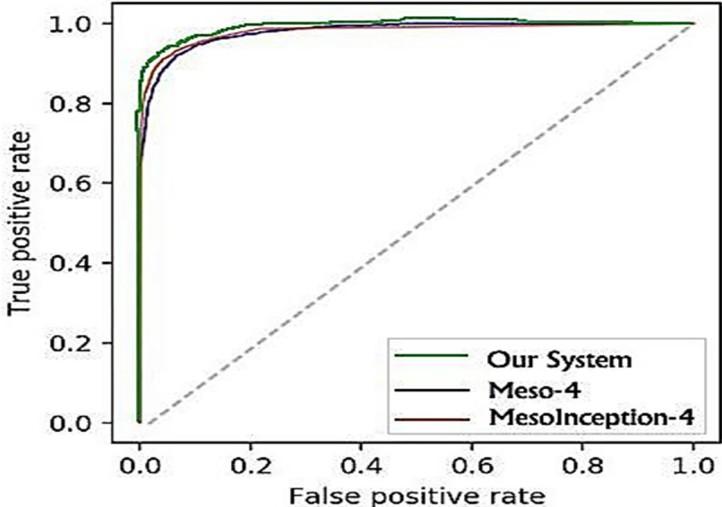

**Figure 11.** ROC curves of the evaluated classifiers on the Face2Face dataset.

The result of the experiment shown in Table 6 indicates that image aggregation improves the detection rates significantly. MesoInception-4 recorded an aggregate higher than 98% and so our system network uses the DeepFake dataset. However, the same score was reached on the Face2Face dataset but left a different aggregate for misclassified videos.

**Table 6.** Video classification scores using image aggregation, with the DeepFake and Face2Face dataset compressed at rate 23.

| Model | Aggregation Score | |
|---|---|---|
| Dataset | DeepFake | Face2Face (23) |
| Meso-4 | 0.969 | 0.953 |
| MesoInception-4 | 0.984 | 0.953 |
| Our System | 0.986 | 0.953 |

To demonstrate the effect of compression on DeepFake detection, image aggregation was also conducted on the intra-frame video compression, this was done to understand if compression rates affect the classification score. This displayed a slightly negative effect on the classification as shown in Table 7. The confusion matrix is displayed in Figure 12.

**Table 7.** I-frames classification score variation on the DeepFake dataset.

| Model | I-Aggregation Score | Difference |
|---|---|---|
| Meso-4 | 0.932 | −0.037 |
| MesoInception-4 | 0.959 | −0.025 |
| Our System | 0.961 | −0.023 |

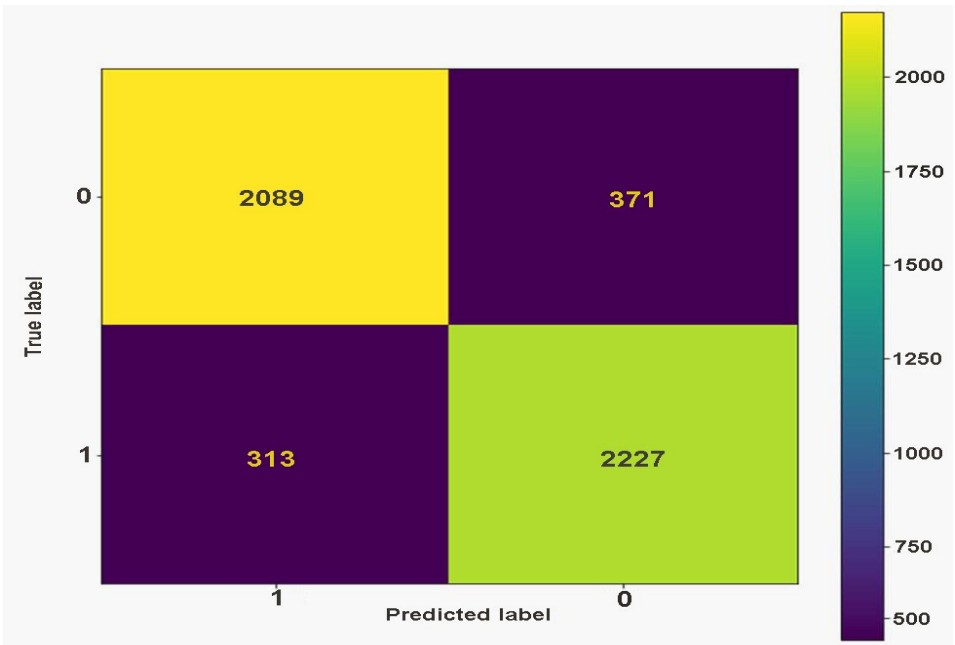

**Figure 12.** Confusion matrix of our system (tn = 2089, fp = 371, fn = 313 tp = 2227).

The comparison of the proposed model with other cutting-edge state-of-the-art models were presented in Table 8 and Figure 13. The proposed model outperforms other models in terms of accuracy and F1-Measure with 0.8632 and 0.8669, respectively. The authors of [9] have a better performance in terms of recall metric with 0.9723, while the authors of [54] have a higher performance in terms of precision with 0.8724. Overall, the proposed model performs better when compared with other baseline models. This shows that the proposed CNN + ReLU model can be used to classify a DeepFake video with better accuracy.

**Table 8.** Performance evaluation in contrast to other cutting-edge techniques in DeepFake detection.

| Authors | Model | Accuracy | Precision | Recall | F1-Measure |
|---|---|---|---|---|---|
| Afchar et al. (2018) [9] | Meso4 | 0.4315 | 0.3580 | 0.9723 | 0.5991 |
| Afchar et al. (2018) [9] | MesoInception4 | 0.7788 | 0.7972 | 0.8143 | 0.8056 |
| Chollet (2017) [54] | Xception | 0.7306 | 0.7973 | 0.6993 | 0.7451 |
| Tan and Le (2019) [55] | EfficientNet-B0 | 0.5964 | 0.7310 | 0.4483 | 0.5558 |
| Sanderson and Lovell (2009) [56] | VGG16 | 0.8103 | 0.8724 | 0.7750 | 0.8208 |
| Our system | CNN + ReLU | 0.8632 | 0.8572 | 0.8768 | 0.8669 |

Since the majority of the inputs are close to zero, the trained CNN is probably susceptible to jitter. A better and more reliable randomly translational non-linear activation for deep CNN can be proposed to address this issue. The use of a hyper-parameter will also enhance the CNN model to be able to select the most appropriate parameter.

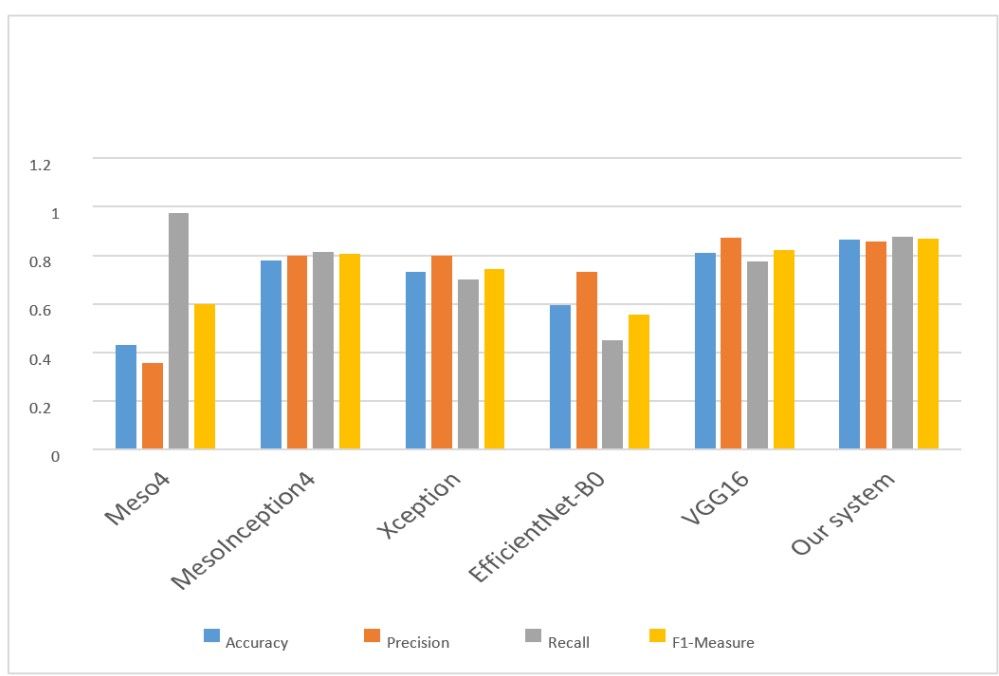

**Figure 13.** Performance evaluation comparison.

## 5. Conclusions and Future Work

Nowadays, the downside of facial manipulation in videos is widely becoming a general concern. This study has extensively analyzed some of the outstanding literature in this field to understand the problem better to propose a network architecture that makes sure such manipulations can be detected using five convolutional neural networks and a recurrent neural network effectively while having a low computational cost. An innovative technique for identifying DeepFakes is presented in this study. The CNN face detector is used in this approach to extract face regions from video frames. The discriminant spatial features of these faces are extracted using ReLU with CNN, assisting in the identification of visual artifacts present in the video frames. Under real-world internet propagation scenarios, this study's technology has an average detection rate of 98% for DeepFake movies and 95% for Face2Face videos, according to the findings. This has greatly shown that the CNN can be enhanced by adding a convolutional layer and other defined parameters. This method also takes into consideration the compression factor which hinders a lot of DeepFake detection mechanisms. More algorithms are expected to develop in the future that will focus more on these aspects while also leveraging updated datasets. The scope of this study has been limited to identifying DeepFakes in still images and videos, but we also believe this method can be extended to detecting DeepFakes in audio and texts and serve as a means to curbing misinformation in this digital age, and these would be investigated in our future work.

**Author Contributions:** The manuscript was written through the contributions of all authors. Conceptualization, J.B.A. and A.T.A.; methodology, J.B.A., A.T.A., R.G.J. and A.L.I.; software, J.B.A. and A.T.A.; validation, A.T.A., J.B.A., A.L.I., C.-T.L., C.-C.L.; formal analysis, A.L.I.; investigation, J.B.A.; resources, A.T.A.; data curation, A.T.A.; writing—original draft preparation, R.G.J.; writing— review and editing, A.T.A., J.B.A., A.L.I., C.-T.L., C.-C.L.; visualization, A.L.I.; supervision, J.B.A.; project administration, A.T.A., J.B.A., A.L.I., C.-T.L., C.-C.L.; funding acquisition, J.B.A. All authors have read and agreed to the published version of the manuscript.

**Funding:** This work was supported by the National Science and Technology Council, Taiwan, R.O.C., under contract no.: MOST 110-2410-H-165-001-MY2.

**Data Availability Statement:** The dataset used is publicly available at https://study.unsw.edu.au/projects/unsw-nb15-dataset (accessed on 14 March 2022).

**Acknowledgments:** The work of Agbotiname Lucky Imoize is supported in part by the Nigerian Petroleum Technology Development Fund (PTDF) and in part by the German Academic Exchange Service (DAAD) through the Nigerian–German Postgraduate Program under grant 57473408.

**Conflicts of Interest:** The authors declare no conflict of interest.

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
