# Peer review of "An Enhanced Deep Learning-Based DeepFake Video Detection and Classification System"

_electronics, doi:10.3390/electronics12010087_

Round 1

Reviewer 1 Report

The paper proposes a DeepFake video detection system using a deep learning approach. Analyzing the paper, I identified the following issues:

1.       The keywords (lines 41-43) need to be rethought. There is a duplicate (“Generative Adversarial Networks”) and some other keywords are somehow redundant (“DeepFake Video Detection” and “Deep Fake Detection” or the triplet “Convolutional Neural Networks”, “Recurrent Neural Networks”, “Generative Adversarial Network”).

2.       English needs polishing. There are grammar mistakes and typos. Some examples: a) line 24: it is written “is” instead of “are”; b) line 36: there is a comma instead of full stop; c) Table 1, last raw on page 7: it is written “Et. al (2019) [28]” instead of “Monti et al (2019) [28]; Please check the entire manuscript.

3.       The sequence “have just been discovered” (line 50) is confusing for something that was first time presented 9 years ago; 

4.       Line 60: the sequence “(GANs)” is inappropriate in the context;

5.       Line 77 (“1.1. Motivation”) is useless in the context and may be deleted;

6.       Either Figure 1 or Figure 2 needs to be deleted. In my view Figure 2, with a better selected title (for example “Forged video creation process using GAN architecture”) is enough. Moreover, presenting the GAN architecture does not match with the purpose of Introduction section.

7.       The information presented in lines 75-137 is basically a Related Work part before section 2 Related Work. I suggest to include this information in Section 2 where it belongs and write a new paragraph instead (in Introduction section) to briefly describe the research gap that was cover.

8.       Lines 142-143: the method was developed only to work on two datasets? It is not a general method to solve DeepFake problems in video data? Please reshape the first contribution;

9.       Figures 4 and 5 are not entirely visible in the version of the manuscript I received. Moreover, Figure 4 is of low quality.

10.   Lines 438-439. The sentence “It turned …” seems to be out of the context;

11.   Lines 559-561 are confusing. For example, TP is the number of records classified as true positive and not “a true positive”. Which is the reason to include the lines 562-565? The entire part from lines 559 to 565 needs to be rewritten;

12.   Table 5: the row that includes “Compression Level” is wrongly included there; Moreover, which is the difference between the two rows containing “ Afchar et al. (2018) [7]”; The authors need to be more precise in describing the two architectures. The same happens in Table 6: “Database” is not a “Network” (see the head of the table and first row); also, the mentioned problem with “Afchar et al. (2018) [7]”; Please reshape.

13.   The quality of Figures 11 and 12 is low.

14.   What exactly “Difference” represents in Table 7.

15.   Please check the format for the references included in References section.

Author Response

Dear Respected Reviewer,

Please see attached file. Thanks for your editorial effort.

Best regards,

Chun-Ta Li

Reviewer 2 Report

The authors present An Enhanced Deep Learning-Based DeepFake Video Detection and Classification System. The study is interesting.  In general, the main conclusions presented in the paper are supported by the figures and supporting text. However, to meet the journal quality standards, the following comments need to be addressed.

•           Abstract: Should be improved and extended. The authors talk lot about the problem formulation, but novelty of the proposed model is missing. Also provided the general applicability of their model. Please be specific what are the main quantitative results to attract general audiences.

•           The introduction can be improved. The authors should focus on extending the novelty of the current study. Emphasize should be given in improvement of the  model (in quantitative  sense)  compared to   existing  state-of-the art models.

•           More details about network architecture and complexity of the model should be provided.

•           what about comparison of the result with current state-of-the art models?  Did authors perform ablation study to compare with different models?

•           What are the baseline models and benchmark results? The authors may compared the result with existing models evaluated with datasets

•           Conclusion parts needs to be strengthened.

•           Please provide a fair weakness and limitation of the model, and how it can be improved.

•           Typographical errors: There are several minor grammatical errors and incorrect sentence structures. Please run this through a spell checker.

·       Discussions of relevant literature could be further enhanced, which can help better motivate the current study and link to the existing work, for example,  relevant deep learning references on object detections   ( see : Neural Networks 2022 https://doi.org/10.1016/j.neunet.2022.05.024; Ecol. Informatics 2022 https://doi.org/10.1016/j.ecoinf.2022.101919; arXiv (2022)  https://doi.org/10.48550/arXiv.2210.04252). Hence they should be briefly discussed in the related work section.

Author Response

(The authors gave the same response as above.)

Round 2

Reviewer 1 Report

The authors have successfully solved all my comments and concerns.

Author Response

Dear Respected Reviewer,

Thanks for your comments.

Best regards,

Chun-Ta Li

Reviewer 2 Report

Although the authors addressed most of the reviewer's previous comments satisfactorily, however, some of the previous points, in particular, comments #2, 3, and 8 needed to incorporate. The authors mentioned that comment #3 has been included in the revised manuscript, but it is still missing. Also the authors should incorporated the relevant literatures suggested by the reviewers in the previous comments. Authors should heiglight all the necessary changes in the revised manuscript. Hence, the current submission is not suitable for publication and needs further revision. After carefully incorporating all previous comments, the manuscript can be considered to proceed further. 

Author Response

Dear Respected Reviewer,

Please see attached file. Thanks for your effort.

Best regards,

Chun-Ta Li
